## [Peer Review File · Nature Microbiology]

Phenogenomics reveals the ecology and evolution of Trichoderma fungi for sustainable agriculture

Corresponding Author: Professor Irina S. Druzhinina

Version 0:

Reviewer comments:

Reviewer #1

(Remarks to the Author)

The manuscript by Steindorff and colleagues reports on a survey of genomic and phenotypic data across 34 species representing the genus *Trichoderma*. This formidable effort to link genomic and phenotypic variation across a range of closely related species of economically important fungi is highly commendable. However, the language used to describe the proposed conceptual framework is confusing and needs to be refined.

In particular, the authors seem to introduce new terms “phenogenes”, which they define as genes linked to fitness-related phenotypes (line 83) and “phenotype-associated homologous orthologous groups, (phenoHOGs)” that are statistically significantly correlated with distinct phenotypic traits (line 373). I am baffled by these definitions and the casual way they are presented. I am familiar with the term “orthologous phenotypes”, or “phenologs”, which predict unique genes associated with specific phenotypes and reveal functionally coherent, evolutionarily conserved gene networks (McGary 2010, www.pnas.org/cgi/doi/10.1073/pnas.0910200107). From the presented narrative, I gather that the authors attempt to introduce their new terms to identify sets of genes that underlie phenologs. If so, this effort requires further development.

I also have a problem with comments on character displacement, which do not seem to have any geographic context, as no data on geographic distribution of focal species are presented (lines 84-86 and 318-327).

Lesser problems:

Line 113. Please include missing citations.

Line 158. Please state the range of gene sizes in addition to referring to Extended Data Figure 1.

Lines 170-175. Where in Fig 1 is this information about results of the gene coevolution network analysis displayed?

Lines 176-181. I appreciate this conclusion. However, I wonder whether it is fully justified, given that it is based on only 34 out of 485 species validly described in the *Trichoderma* genus (line 103).

Line 197-201. These conclusions seem highly speculative. I would leave them for the Discussion section.

Line 241-245. Please include a citation for info on sialic acids.

Line 338-340. Please specify enzymatic abilities underpinning these traits.

Line 408. What is a “postgenomic toolbox”?

Line 428. Please correct a typo in “expansins”.

Fig 1. Cartoon lifestyle identifiers are too complex and difficult to distinguish. Please simplify. It is difficult to distinguish the lifestyle legend from the phylogeny. Please differentiate.

Fig 3a. “Loinear”?

Fig 3b. Please explain abbreviations in the caption.

Extended Data Figure 2. Please explain in the caption the meaning of “germination profile” and the scale from -2 to 8.

Reviewer #2

(Remarks to the Author)

Introduction:

Biotrophic *Trichoderma* strains can also directly control plant pathogenic fungi and nematodes, positively affecting crop microbiomes- Needs some clarification here. The genus is primarily saprophyte, sometimes behaving as hemibiotrophs while attacking other fungi. Here the authors claim that some are biotrophs. Please discuss this aspect elaborately, will be good for readers to get a clear idea about their existence in nature- as saprophyte, hemibiotroph or biotroph, or all of these!

Several species can also become endophytic in roots, stems and leaves of wooden plants(refs)- please include references, seems like an unedited version!

Additionally, reports of crop pathogenicity of *Trichoderma* become common- or should it be becoming common?

Results:

The optimal carbon source for all *Trichoderma* strains was N-acetyl-D-glucosamine, a monomer of chitin found in fungal cell walls and the exoskeletons of arthropods. This preference underscores mycoparasitic capabilities of *Trichoderma*, its competitive interactions within fungal communities and occasional parasitism on insects- many *Trichoderma* are aggressive parasite on oomycetes (lacking chitin in their cell wall), please through some light on this aspect.

Limited growth on a number of specialized substrates reveals ecological conditions likely unfavorable for *Trichoderma* colonization, possibly characterized by anaerobic environments- if I understood correctly, the authors are emphasizing that phyllosphere is the main habitat of *Trichoderma*. Phyllosphere does not really represent an anaerobic environment, or does it? This point needs to be discussed for better clarity.

Genotypic Diversity and Nutritional Opportunities Influence Reproductive Strategies of *Trichoderma* strains: No discussion on chlamydospores

Negligible Antibacterial Activity: Not really universal, some metabolites, like gliotoxin, exhibit strong antimicrobial activity.

Discussion:

..perfectly adapted for mycoparasitism within phyllosphere-associated microbial biofilms (bark, leaves, roots)..- this is very interesting and novel insight. However, do we consider roots as part of phyllosphere?

..robust but non-circadian response to light..- Are the authors contradicting such studies: "Circadian oscillations in *Trichoderma atroviride* and the role of core clock components in secondary metabolism, development, and mycoparasitism against the phytopathogen *Botrytis cinerea*. Henríquez-Urrutia M, Spanner R, Olivares-Yáñez C, Seguel-Avello A, Pérez-Lara R, Guillén-Alonso H, Winkler R, Herrera-Estrella A, Canessa P, Larrondo LF. *Elife*. 2022 Aug 11;11:e71358. doi: 10.7554/eLife.71358", if so please indicate clearly.

Unfortunately, testing the hypothesis that *Trichoderma* is predominantly associated with tropical rainforest canopies and becomes edaphic in temperate or polar regions is currently unfeasible due to a sampling bias,...- towards the end of the manuscript, this is a confusing statement (at least I'm confused), with previous discussion on phyllosphere as the primary habitat; this can be put in a better way.

Additionally, the proposed affinity to the phyllosphere points to biotrophic associations with plants- is this true? Are they really biotrophs with no saprophytic existence?

General:

Well defined strain variability, such as those existing in *T. virens*, not accounted for.

Secondary metabolism, especially presence of known mycotoxins (a significant risk to safe use), has not been studied/discussed.

Phenotypes that have been published already should have been taken into account. The authors' aim was to "provide an evolutionary framework for understanding fitness trade-offs between environmental opportunism and ecological specialization". However, discussion on why some species/strains are human pathogens, a very important aspect related to safety, is lacking. Similarly, can we predict if a particular strain has the potential to be a plant pathogen? This is very important as it's not possible to assess the behaviour on all plants before commercial release of a formulation. The discussion begins with "The aim of our study was to utilize a postgenomic toolbox to provide understanding of *Trichoderma* ecology and evolution, which underpin the science-based risk assessment of its application in sustainable agriculture", however, at the end of the day, do we have such a toolbox using which we can assess the risks?

Some *Trichoderma* (like *T. virens*) spores are embedded in gelatinous matrix and hence not air-borne, while most others are produce dry spores which are easily air-borne. Did the authors look for such phenotype in the strains under study?

Reviewer #3

(Remarks to the Author)

This manuscript offers a comprehensive insight into *Trichoderma* and overall the work is of high quality, with good and clear writing and was very pleasant to read. I have a selection of major and minor comments, which could help improve further this manuscript. But first, I would like to address a critical concern regarding data and algorithm availability. In the corresponding sections (data availability and code availability), the authors wrote that 'additional data supporting the findings are available from the corresponding author upon reasonable request' (line 519-521) and 'The REPAINT AI algorithm (...) is available from the corresponding author upon reasonable request' (line 523/534). The withholding of data and algorithm without clear and adequate reasons is not acceptable and not compatible with high quality work put together in this manuscript. Gatekeeping practices do not offer any benefits to the community and can only lead to loss of data and reproducibility in the long term, as highlighted in this paper: doi.org/10.1038/s41597-021-00981-0.

List of major comments:

Result 'Ecological Adaptations in *Trichoderma* Genomes Enhanced by Regulatory Genes, Small Secreted Proteins, and Genes of Unknown Function'. This section is rather confusing to read (see comments later about the associated methods section). I cannot find the "more than 150 parameters" in the Supplementary Data 2. Or does it correspond to Supplementary Data 2D? In which case the number of parameters is 141? Does the "uncertainty of each trait" (line 372) means that the authors were not able to test for the traits, or that the results were not conclusive?

Figure 2: the regression lines don't add any value to the figure, except showing that they are a poor fit. And they are not used in the main text. The PCA plot needs to have clear axes (Principal component 1, principal component 2) and the amount of variation they explain. The coloring in the PCA is also confusing with some strain having two colors, please clarify why. Overall, the use of average growth rate across all carbon sources is misleading and hides the variation of growth rate per strain across multiple carbon sources.

Figure 3b. I don't see where that figure is used/mentioned in the text. Figure 3 is referenced at line 291, 296, 314, 330, 333 but only for information regarding Fig 3a. Make sure Figure 3b is referenced and used in the main text, or consider removing it from the Figure. Also, the acronyms (GT, GH, CE, PL, SM, TC, NRPS, PKS) used in Fig 3b are not explained in the legend (or in the text).

Figure 5: Panel C and D are not used or referenced in the main text (panel D corresponds to the last sentence in the result section). Consider removing them, or discuss their relevance in the main text. Panel C is hardly readable, both from the multiple lines linking phenotypes, to the phenotype colors (circle line too thin). An alternative could be a heatmap-like figure. In panel B, some transcription factors are found only in phenogenes (asterix mark) but not in the genomes? It raises the question of their origin. Or did the author compute the TF found in phenogenes, and the ones found in the REST of the genomes? Please clarify this in the figure and legend.

Methods 'Genome sequencing an assembly'. Line 1024-1025 "An automated attempt was made to reassemble any potential organelle (mitochondrion) from the filtered reads and remove any organelle-matching reads with kmer matching against the resulting contigs with an in-house tool.". Code should be available for full reproducibility.

Methods "genomic feature selection". Important information is missing: it is not clear what is the training dataset and the tested dataset (genomes and phenotypes), what kind of kernel was used (linear?) and how the correlation values were obtained (Pearson?). Please add the version of scikit-learn used, as "sklearn.cross_validation" is not part of scikit-learn since 2016 (v0.18). Overall, it would be better to provide the full code (github repository, zenodo, figshare, etc) for clarity and reproducibility of the results.

List of minor comments:

Line 113: reference missing (noted "(refs)")

Line 123-124, "...the identification of phenogenes - genes enhancing phenetic traits": as the authors define phenogenes later (line 130), these genes are associated with phenotypes, rather than 'enhancing' them.

Figure 3a. The last column "product" is not defined, but I am assuming it is the product of the three fitness related-trait. It would be nice if clarified in the legend. Similarly, I am assuming that the stars represent cold and drought resistant strains, but their annotation is not consistent (lacking for the first two stars) and could be described in the legend.

Figure 4a: Why is the phenotype associated represented in a two dimensional plot when it is a one dimension value (prob of 1 = 1 - prob of 0). Clarity could be improved with a simple vertical dot plot or something similar.

Methods: missing software version for AlphaFold2, Foldseek, PhyKIT, scikit-learn.

Reviewer #4

(Remarks to the Author)

This is an ambitious and comprehensive study that combines genomics, phenotyping, and ecological profiling of *Trichoderma* spp. to understand their ecological versatility and to help assess their implications in terms of agricultural applications and also the risk that they pose to crops and animals. The study is based on the premise that while *Trichoderma* species hold enormous promise as biofungicides and growth-promoting agents, and as they are increasingly deployed across fields and crops, their evolutionary plasticity and occasional pathogenicity require careful risk assessment. This is a strong premise and makes a strong argument for the proposed work. And the work reported make a valuable contribution to fungal ecology and evolutionary biology; but the manuscript in its current feels unfocussed and the main core of the paper is lost in the abundance of the data. Fig 3 contains a donating amount of information and it is not clear whether or not all of it is required. The manuscript is dense and at times overloaded with detail, which can obscure the main conclusions. In particular, the link between the work reported and the biosafety risk is weak and almost appears as an afterthought, after the first paragraph of the introduction. None of the figures, it seems, contribute to that question of biosafety risk, or at least, not clearly. A more streamlined version, possibly emphasizing ecological and evolutionary implications, with a tighter focus and more concise presentation would improve clarity and make for a more impactful contribution. In particular, if the focus is to be on balancing biosafety risks, as introduced at the beginning of the paper, via an evolutionary framework, there needs to be a better narrative. Alternatively, the authors could just drop the biosafety risk claims and just focus on the ecology and evolution of the group. Finally, the manuscript is currently too long, feels meandering, and also contains multiple typos that should be fixed prior to a new submission. The authors can be congratulated for having produced a very extensive dataset and I am sure that a more streamlined version of their work will be impactful.

Decision Letter:

31st July 2025

Dear Professor Druzhinina,

Thank you for your patience while your manuscript "Ecological Genomic Research in *Trichoderma* Unveils Risks and Opportunities for Its Use in Sustainable Agriculture" was under peer-review at Nature Microbiology. Please accept my apologies for the delay in our decision. It has now been seen by 4 referees, whose expertise and comments you will find at the end of this email. Although they find your work of some potential interest, they have raised a number of concerns that will need to be addressed before we can consider publication of the work in Nature Microbiology.

In particular, we suggest that you make major changes to the manuscript to address the technical concerns with the computational methods, improve the claims made regarding the implication of this work in sustainable agriculture, address the conflicts highlighted by the referees with the current literature, improve the code/data availability, and provide more context with improved discussion. Should this allow you to address these criticisms, we would be happy to look at a revised manuscript.

Please include a data availability statement as a separate section after Methods but before references, under the heading "Data Availability". This section should inform readers about the availability of the data used to support the conclusions of your study. This information includes accession codes to public repositories (data banks for protein, DNA or RNA sequences, microarray, proteomics data etc...), references to source data published alongside the paper, unique identifiers such as URLs to data repository entries, or data set DOIs, and any other statement about data availability. At a minimum, you should include the following statement: "The data that support the findings of this study are available from the corresponding author upon request", mentioning any restrictions on availability. If DOIs are provided, we also strongly encourage including these in the Reference list (authors, title, publisher (repository name), identifier, year). For more guidance on how to write this section please see: <http://www.nature.com/authors/policies/data/data-availability-statements-data-citations.pdf>

* If you have not done so already we suggest that you begin to revise your manuscript so that it conforms to our Article format instructions at <http://www.nature.com/nmicrobiol/info/final-submission>. Refer also to any guidelines provided in this letter.

When submitting the revised version of your manuscript, please pay close attention to our [href="https://www.nature.com/nature-portfolio/editorial-policies/image-integrity">Digital Image Integrity Guidelines.](https://www.nature.com/nature-portfolio/editorial-policies/image-integrity) and to the following points below:

EXTENDED DATA FIGURES

Link Redacted

Note: This url links to your confidential homepage and associated information about manuscripts you may have submitted or be reviewing for us. If you wish to forward this e-mail to co-authors, please delete this link to your homepage first.

Nature Microbiology is committed to improving transparency in authorship. As part of our efforts in this direction, we are now requesting that all authors identified as 'corresponding author' on published papers create and link their Open Researcher and Contributor Identifier (ORCID) with their account on the Manuscript Tracking System (MTS), prior to acceptance. This applies to primary research papers only. ORCID helps the scientific community achieve unambiguous attribution of all scholarly contributions. You can create and link your ORCID from the home page of the MTS by clicking on 'Modify my Springer Nature account'. For more information please visit [please visit www.springernature.com/orcid](http://www.springernature.com/orcid).

If you wish to submit a suitably revised manuscript we would hope to receive it within 6 months. If you cannot send it within this time, please let us know. We will be happy to consider your revision, even if a similar study has been accepted for publication at Nature Microbiology or published elsewhere (up to a maximum of 6 months).

Yours sincerely,

Reviewer Expertise:

- Referee #1: Fungal evolutionary biology
- Referee #2: Trichoderma
- Referee #3: Machine learning and genomics
- Referee #4: Fungal genomics

Reviewer Comments:

Reviewer #1 (Remarks to the Author):

The manuscript by Steindorff and colleagues reports on a survey of genomic and phenotypic data across 34 species representing the genus *Trichoderma*. This formidable effort to link genomic and phenotypic variation across a range of closely related species of economically important fungi is highly commendable. However, the language used to describe the proposed

conceptual framework is confusing and needs to be refined.

In particular, the authors seem to introduce new terms “phenogenes”, which they define as genes linked to fitness-related phenotypes (line 83) and “phenotype-associated homologous orthologous groups, (phenoHOGs)” that are statistically significantly correlated with distinct phenotypic traits (line 373). I am baffled by these definitions and the casual way they are presented. I am familiar with the term “orthologous phenotypes”, or “phenologs”, which predict unique genes associated with specific phenotypes and reveal functionally coherent, evolutionarily conserved gene networks (McGary 2010, www.pnas.org/cgi/doi/10.1073/pnas.0910200107). From the presented narrative, I gather that the authors attempt to introduce their new terms to identify sets of genes that underlie phenologs. If so, this effort requires further development.

I also have a problem with comments on character displacement, which do not seem to have any geographic context, as no data on geographic distribution of focal species are presented (lines 84-86 and 318-327).

Lesser problems:

Line 113. Please include missing citations.

Line 158. Please state the range of gene sizes in addition to referring to Extended Data Figure 1.

Lines 170-175. Where in Fig 1 is this information about results of the gene coevolution network analysis displayed?

Lines 176-181. I appreciate this conclusion. However, I wonder whether it is fully justified, given that it is based on only 34 out of 485 species validly described in the *Trichoderma* genus (line 103).

Line 197-201. These conclusions seem highly speculative. I would leave them for the Discussion section.

Line 241-245. Please include a citation for info on sialic acids.

Line 338-340. Please specify enzymatic abilities underpinning these traits.

Line 408. What is a “postgenomic toolbox”?

Line 428. Please correct a typo in “expansins”.

Fig 1. Cartoon lifestyle identifiers are too complex and difficult to distinguish. Please simplify. It is difficult to distinguish the lifestyle legend from the phylogeny. Please differentiate.

Fig 3a. “Loinear”?

Fig 3b. Please explain abbreviations in the caption.

Extended Data Figure 2. Please explain in the caption the meaning of “germination profile” and the scale from -2 to 8.

Reviewer #2 (Remarks to the Author):

Introduction:

Biotrophic *Trichoderma* strains can also directly control plant pathogenic fungi and nematodes, positively affecting crop microbiomes- Needs some clarification here. The genus is primarily saprophyte, sometimes behaving as hemibiotrophs while attacking other fungi. Here the authors claim that some are biotrophs. Please discuss this aspect elaborately, will be good for readers to get a clear idea about their existence in nature- as saprophyte, hemibiotroph or biotroph, or all of these!

Several species can also become endophytic in roots, stems and leaves of wooden plants(refs)- please include references, seems like an unedited version!

Additionally, reports of crop pathogenicity of *Trichoderma* become common- or should it be becoming common?

Results:

The optimal carbon source for all *Trichoderma* strains was N-acetyl-D-glucosamine, a monomer of chitin found in fungal cell walls and the exoskeletons of arthropods. This preference underscores mycoparasitic capabilities of *Trichoderma*, its competitive interactions within fungal communities and occasional parasitism on insects- many *Trichoderma* are aggressive parasite on oomycetes (lacking chitin in their cell wall), please through some light on this aspect.

Limited growth on a number of specialized substrates reveals ecological conditions likely unfavorable for *Trichoderma* colonization, possibly characterized by anaerobic environments- if I understood correctly, the authors are

emphasizing that phyllosphere is the main habitat of *Trichoderma*. Phyllosphere does not really represent an anaerobic environment, or does it? This point needs to be discussed for better clarity.

Genotypic Diversity and Nutritional Opportunities Influence Reproductive Strategies of *Trichoderma* strains: No discussion on chlamydospores

Negligible Antibacterial Activity: Not really universal, some metabolites, like gliotoxin, exhibit strong antimicrobial activity.

Discussion:

..perfectly adapted for mycoparasitism within phyllosphere-associated microbial biofilms (bark, leaves, roots)..- this is very interesting and novel insight. However, do we consider roots as part of phyllosphere?

..robust but non-circadian response to light.- Are the authors contradicting such studies: "Circadian oscillations in *Trichoderma atroviride* and the role of core clock components in secondary metabolism, development, and mycoparasitism against the phytopathogen *Botrytis cinerea*. Henríquez-Urrutia M, Spanner R, Olivares-Yáñez C, Seguel-Avello A, Pérez-Lara R, Guillén-Alonso H, Winkler R, Herrera-Estrella A, Canessa P, Larrondo LF. *Elife*. 2022 Aug 11;11:e71358. doi: 10.7554/eLife.71358", if so please indicate clearly.

Unfortunately, testing the hypothesis that *Trichoderma* is predominantly associated with tropical rainforest canopies and becomes edaphic in temperate or polar regions is currently unfeasible due to a sampling bias,...- towards the end of the manuscript, this is a confusing statement (at least I'm confused), with previous discussion on phyllosphere as the primary habitat; this can be put in a better way.

Additionally, the proposed affinity to the phyllosphere points to biotrophic associations with plants- is this true? Are they really biotrophs with no saprophytic existence?

General:

Well defined strain variability, such as those existing in *T. virens*, not accounted for.

Secondary metabolism, especially presence of known mycotoxins (a significant risk to safe use), has not been studied/discussed.

Phenotypes that have been published already should have been taken into account. The authors' aim was to "provide an evolutionary framework for understanding fitness trade-offs between environmental opportunism and ecological specialization". However, discussion on why some species/strains are human pathogens, a very important aspect related to safety, is lacking. Similarly, can we predict if a particular strain has the potential to be a plant pathogen? This is very important as it's not possible to assess the behaviour on all plants before commercial release of a formulation. The discussion begins with "The aim of our study was to utilize a postgenomic toolbox to provide understanding of *Trichoderma* ecology and evolution, which underpin the science-based risk assessment of its application in sustainable agriculture", however, at the end of the day, do we have such a toolbox using which we can assess the risks?

Some *Trichoderma* (like *T. virens*) spores are embedded in gelatinous matrix and hence not air-borne, while most others are produce dry spores which are easily air-borne. Did the authors look for such phenotype in the strains under study?

Reviewer #3 (Remarks to the Author):

This manuscript offers a comprehensive insight into *Trichoderma* and overall the work is of high quality, with good and clear writing and was very pleasant to read. I have a selection of major and minor comments, which could help improve further this manuscript. But first, I would like to address a critical concern regarding data and algorithm availability. In the corresponding sections (data availability and code availability), the authors wrote that 'additional data supporting the findings are available from the corresponding author upon reasonable request' (line 519-521) and 'The REPAINT AI algorithm (...) is available from the corresponding author upon reasonable request' (line 523/534). The withholding of data and algorithm without clear and adequate reasons is not acceptable and not compatible with high quality work put together in this manuscript. Gatekeeping practices do not offer any benefits to the community and can only lead to loss of data and reproducibility in the long term, as highlighted in this paper: doi.org/10.1038/s41597-021-00981-0.

List of major comments:

Result 'Ecological Adaptations in *Trichoderma* Genomes Enhanced by Regulatory Genes, Small Secreted Proteins, and Genes of Unknown Function'. This section is rather confusing to read (see comments later about the associated methods section). I cannot find the "more than 150 parameters" in the Supplementary Data 2. Or does it correspond to Supplementary Data 2D? In which case the number of parameters is 141? Does the "uncertainty of each trait" (line 372) means that the authors were not able to test for the traits, or that the results were not conclusive?

Figure 2: the regression lines don't add any value to the figure, except showing that they are a poor fit. And they are not used in

the main text. The PCA plot needs to have clear axes (Principal component 1, principal component 2) and the amount of variation they explain. The coloring in the PCA is also confusing with some strain having two colors, please clarify why. Overall, the use of average growth rate across all carbon sources is misleading and hides the variation of growth rate per strain across multiple carbon sources.

Figure 3b. I don't see where that figure is used/mentioned in the text. Figure 3 is referenced at line 291, 296, 314, 330, 333 but only for information regarding Fig 3a. Make sure Figure 3b is referenced and used in the main text, or consider removing it from the Figure. Also, the acronyms (GT, GH, CE, PL, SM, TC, NRPS, PKS) used in Fig 3b are not explained in the legend (or in the text).

Figure 5: Panel C and D are not used or referenced in the main text (panel D corresponds to the last sentence in the result section). Consider removing them, or discuss their relevance in the main text. Panel C is hardly readable, both from the multiple lines linking phenotypes, to the phenotype colors (circle line too thin). An alternative could be a heatmap-like figure. In panel B, some transcription factors are found only in phenogenes (asterix mark) but not in the genomes? It raises the question of their origin. Or did the author compute the TF found in phenogenes, and the ones found in the REST of the genomes? Please clarify this in the figure and legend.

Methods 'Genome sequencing an assembly'. Line 1024-1025 "An automated attempt was made to reassemble any potential organelle (mitochondrion) from the filtered reads and remove any organelle-matching reads with kmer matching against the resulting contigs with an in-house tool.". Code should be available for full reproducibility.

Methods "genomic feature selection". Important information is missing: it is not clear what is the training dataset and the tested dataset (genomes and phenotypes), what kind of kernel was used (linear?) and how the correlation values were obtained (Pearson?). Please add the version of scikit-learn used, as "sklearn.cross_validation" is not part of scikit-learn since 2016 (v0.18). Overall, it would be better to provide the full code (github repository, zenodo, figshare, etc) for clarity and reproducibility of the results.

List of minor comments:

Line 113: reference missing (noted "(refs)")

Line 123-124, "...the identification of phenogenes - genes enhancing phenetic traits": as the authors define phenogenes later (line 130), these genes are associated with phenotypes, rather than 'enhancing' them.

Figure 3a. The last column "product" is not defined, but I am assuming it is the product of the three fitness related-trait. It would be nice if clarified in the legend. Similarly, I am assuming that the stars represent cold and drought resistant strains, but their annotation is not consistent (lacking for the first two stars) and could be described in the legend.

Figure 4a: Why is the phenotype associated represented in a two dimensional plot when it is a one dimension value (prob of 1 = 1 - prob of 0). Clarity could be improved with a simple vertical dot plot or something similar.

Methods: missing software version for AlphaFold2, Foldseek, PhyKIT, scikit-learn.

Reviewer #4 (Remarks to the Author):

This is an ambitious and comprehensive study that combines genomics, phenotyping, and ecological profiling of *Trichoderma* spp. to understand their ecological versatility and to help assess their implications in terms of agricultural applications and also the risk that they pose to crops and animals. The study is based on the premise that while *Trichoderma* species hold enormous promise as biofungicides and growth-promoting agents, and as they are increasingly deployed across fields and crops, their evolutionary plasticity and occasional pathogenicity require careful risk assessment. This is a strong premise and makes a strong argument for the proposed work. And the work reported make a valuable contribution to fungal ecology and evolutionary biology; but the manuscript in its current feels unfocussed and the main core of the paper is lost in the abundance of the data. Fig 3 contains a donating amount of information and it is not clear whether or not all of it is required. The manuscript is dense and at times overloaded with detail, which can obscure the main conclusions. In particular, the link between the work reported and the biosafety risk is weak and almost appears as an afterthought, after the first paragraph of the introduction. None of the figures, it seems, contribute to that question of biosafety risk, or at least, not clearly. A more streamlined version, possibly emphasizing ecological and evolutionary implications, with a tighter focus and more concise presentation would improve clarity and make for a more impactful contribution. In particular, if the focus is to be on balancing biosafety risks, as introduced at the beginning of the paper, via an evolutionary framework, there needs to be a better narrative. Alternatively, the authors could just drop the biosafety risk claims and just focus on the ecology and evolution of the group. Finally, the manuscript is currently too long, feels

meandering, and also contains multiple typos that should be fixed prior to a new submission. The authors can be congratulated for having produced a very extensive dataset and I am sure that a more streamlined version of their work will be impactful.

Version 1:

Reviewer comments:

Reviewer #1

(Remarks to the Author)

The revised manuscript by Steindorff and colleagues is much improved, with my original comments addressed satisfactorily. However, some sections and figures could use additional editing to increase clarity and impact.

Line 75 I would say "Results identify *Trichoderma* as an ancient, genetically cohesive..."

Line 133. Please define "HOGs".

Lines 149-164. The section "The Start of *Trichoderma* Life Cycle..." needs an introductory sentence explaining the rationale for the presented experiments and why the phyllosphere microbial mats are important in the first place. The paragraphs seem choppy, disconnected from each other and devoid of a broader synthesis.

Line 170. Please explain the term "idioadaptation".

Line 204. I do not follow the opening comment "To further assess *Trichoderma* fitness..." Do the authors mean "fitness-related traits"?

Lines 300-301. I would say "Ecological Adaptations in *Trichoderma* Genomes Mediated by Regulatory Genes, Small Secreted Proteins, and Genes of Unknown Function."

Line 302. I would say "We demonstrated that *Trichoderma* spp. maintain genomic cohesion while undergoing adaptive radiation and developing significant ecophysiological variability..."

Figure 2. I found panel (a) not very informative and would suggest showing only one species per strategy. The combination of panel (b) and panel (c) is great. However, I do not see the colors mentioned in the caption to be represented in panel (c). Please adjust the diagram in panel (c) to represent these colors.

Figure 3. I love the info content of this figure. However, I would suggest using more colors and shades to denote responses on different substrates in (a). Also, using shades of purple and green for both taxonomy and responses is confusing.

Supplementary Figure 4, which provides support for most of the section "Profound Edaphic Incompetence...", conveys a massive amount of information. Please provide a more detailed and meaningful caption describing this content.

Reviewer #2

(Remarks to the Author)

Reviewer #3

(Remarks to the Author)

I thank the authors for addressing my comments, particularly regarding the accessibility of the code used in to process the data presented in the manuscript. I still have some remarks regarding a few points in the revised manuscript.

I appreciate that the authors have now referenced Fig. 3b and clarified its intended contrast with phenotypic variability in Fig. 3a. However, the integration remains largely descriptive, and the term "genome inventories" overgeneralizes what is shown (specific gene category counts rather than total genomic content). To justify inclusion of panel b in the main figure, the authors should provide a minimal analytical or quantitative link, such as a measure of variance in these genomic features, a comparison of their range relative to phenotypic variation or a statement on whether any of these categories correlate with fitness metrics.

If such analysis is beyond the present scope, I recommend moving panel b to the Extended Data or omitting it altogether, since its current treatment does not meaningfully advance the results or interpretation.

As a minor note, I noticed some inconsistency between the original and revised Figure 1:

T. inhamatum, *T. spirale* and *Clonostachys* have different substrates between the two figures. In the revised version, *T. inhamatum* does not have "soil saprotrophy", *T. spirale* has the symbol for 'endophyte, epiphyte, parasite, rhizosphere colonizer',

which was absent in the first version, and *Clonostachys* has two new symbols for 'mycoparasite' and 'litter decomposition'.

Reviewer #4

(Remarks to the Author)

This manuscript provides a valuable and highly ambitious study, integrating comparative genomics with large-scale ecophysiological, reproductive, stress-tolerance, and biotic-interaction assays across 37 strains. It has the potential to become a landmark reference on *Trichoderma* biology. The study also highlights biosafety concerns regarding opportunistic species, such as those with human or plant pathogenicity, especially in the context of their use as biocontrol agents or biofungicides in agriculture. I congratulate the authors for the extensive analyses conducted and the formidable dataset generated and presented. The authors have now streamlined the narrative, compared to an earlier version, and provided significant clarifications in response to earlier critiques. There is still some overinterpretation in some sections that could be tempered down.

Major comments:

The authors state that contrasting phenotypic profiles among sympatric or closely related taxa indicate character displacement. However, true character displacement would require evidence of niche partitioning driven by interspecific competition within sympatry. Here, sympatry is assumed but not demonstrated (e.g., no biogeographic co-occurrence data). Differences in BIOLOG phenotypes alone are insufficient. Perhaps the claims could be rephrased as "consistent with" or "suggestive of" character displacement.

The authors highlight *T. afroharzianum* and *T. longibrachiatum* as high-concern species from a biosafety perspective, referencing plant and human pathogenicity. However, strain-level variation could be important and should probably be considered instead of making species-level generalizations. Agronomic recommendations rely on regulatory guidance that would focus on a strain that would be submitted to the regulatory process. Species are not regulated, but strains are. I think that the biosafety section is important as it provides a solid application of the findings of this work, but it would be prudent to emphasize that the data could constitute a first-tier screening, not prescriptive regulatory conclusions.

The authors addressed the rest of the comments appropriately in the manuscript and the rebuttal letter.

Decision Letter:

26th November 2025

Dear Professor Druzhinina,

Thank you for your patience while your manuscript "Phenogenomic analysis of *Trichoderma* reveals determinants of fungal fitness" was under peer-review at Nature Microbiology. It has now been seen by 4 referees, whose comments you will find at the of this email. You will see from their comments below that they are happy with the revision but they have also asked for many changes to the text and figures to improve the manuscript. We are very interested in the possibility of publishing your study in Nature Microbiology, so, we would like to consider your response to these concerns in the form of a revised manuscript before we make a final decision on publication.

If you have not done so already please begin to revise your manuscript so that it conforms to our Article format instructions at <http://www.nature.com/nmicrobiol/info/final-submission/>

The usual length limit for a Nature Microbiology Article is six display items (figures or tables) and 3,500 words. We have some flexibility, and can allow a revised manuscript at 4,500 words, but please consider this a firm upper limit. There is a trade-off of ~250 words per display item, so if you need more space, you could move a Figure or Table to Supplementary Information.

Some reduction could be achieved by focusing any introductory material and moving it to the start of your opening 'bold' paragraph, whose function is to outline the background to your work, describe in a sentence your new observations, and explain your main conclusions. The discussion should also be limited. Methods should be described in a separate section following the discussion, we do not place a word limit on Methods.

Nature Microbiology titles should give a sense of the main new findings of a manuscript, and should not contain punctuation. Please keep in mind that we strongly discourage active verbs in titles, and that they should ideally fit within 90 characters each (including spaces).

We strongly support public availability of data. Please place the data used in your paper into a public data repository, if one exists, or alternatively, present the data as Source Data or Supplementary Information. If data can only be shared on request, please explain why in your Data Availability Statement, and also in the correspondence with your editor. For some data types, deposition in a public repository is mandatory - more information on our data deposition policies and available repositories can

be found at <https://www.nature.com/nature-research/editorial-policies/reporting-standards#availability-of-data>.

Please include a data availability statement as a separate section after Methods but before references, under the heading "Data Availability". This section should inform readers about the availability of the data used to support the conclusions of your study. This information includes accession codes to public repositories (data banks for protein, DNA or RNA sequences, microarray, proteomics data etc...), references to source data published alongside the paper, unique identifiers such as URLs to data repository entries, or data set DOIs, and any other statement about data availability. At a minimum, you should include the following statement: "The data that support the findings of this study are available from the corresponding author upon request", mentioning any restrictions on availability. If DOIs are provided, we also strongly encourage including these in the Reference list (authors, title, publisher (repository name), identifier, year). For more guidance on how to write this section please see: <http://www.nature.com/authors/policies/data/data-availability-statements-data-citations.pdf>

To improve the accessibility of your paper to readers from other research areas, please pay particular attention to the wording of the paper's opening bold paragraph, which serves both as an introduction and as a brief, non-technical summary in about 150 words. If, however, you require one or two extra sentences to explain your work clearly, please include them even if the paragraph is over-length as a result. The opening paragraph should not contain references. Because scientists from other sub-disciplines will be interested in your results and their implications, it is important to explain essential but specialised terms concisely. We suggest you show your summary paragraph to colleagues in other fields to uncover any problematic concepts.

If your paper is accepted for publication, we will edit your display items electronically so they conform to our house style and will reproduce clearly in print. If necessary, we will re-size figures to fit single or double column width. If your figures contain several parts, the parts should form a neat rectangle when assembled. Choosing the right electronic format at this stage will speed up the processing of your paper and give the best possible results in print. We would like the figures to be supplied as vector files - EPS, PDF, AI or postscript (PS) file formats (not raster or bitmap files), preferably generated with vector-graphics software (Adobe Illustrator for example). Please try to ensure that all figures are non-flattened and fully editable. All images should be at least 300 dpi resolution (when figures are scaled to approximately the size that they are to be printed at) and in RGB colour format. Please do not submit Jpeg or flattened TIFF files. Please see also 'Guidelines for Electronic Submission of Figures' at the end of this letter for further detail.

Figure legends must provide a brief description of the figure and the symbols used, within 350 words, including definitions of any error bars employed in the figures.

When submitting the revised version of your manuscript, please pay close attention to our [href="https://www.nature.com/nature-research/editorial-policies/image-integrity">Digital Image Integrity Guidelines.](https://www.nature.com/nature-research/editorial-policies/image-integrity) and to the following points below:

EXTENDED DATA FIGURES

Please include a statement before the acknowledgements naming the author to whom correspondence and requests for materials should be addressed.

Finally, we require authors to include a statement of their individual contributions to the paper -- such as experimental work, project planning, data analysis, etc. -- immediately after the acknowledgements. The statement should be short, and refer to authors by their initials. For details please see the Authorship section of our joint Editorial policies at http://www.nature.com/authors/editorial_policies/authorship.html

* include a point-by-point response to any editorial suggestions and to our referees. Please include your response to the editorial suggestions in your cover letter, and please upload your response to the referees as a separate document.

* ensure it complies with our format requirements for Letters as set out in our guide to authors at www.nature.com/nmicrobiol/info/gta/

* state in a cover note the length of the text, methods and legends; the number of references; number and estimated final size of figures and tables

* resubmit electronically if possible using the link below to access your home page:

Link Redacted

*This url links to your confidential homepage and associated information about manuscripts you may have submitted or be reviewing for us. If you wish to forward this e-mail to co-authors, please delete this link to your homepage first.

Please ensure that all correspondence is marked with your Nature Microbiology reference number in the subject line.

Nature Microbiology is committed to improving transparency in authorship. As part of our efforts in this direction, we are now requesting that all authors identified as 'corresponding author' on published papers create and link their Open Researcher and Contributor Identifier (ORCID) with their account on the Manuscript Tracking System (MTS), prior to acceptance. This applies to primary research papers only. ORCID helps the scientific community achieve unambiguous attribution of all scholarly contributions. You can create and link your ORCID from the home page of the MTS by clicking on 'Modify my Springer Nature account'. For more information please visit www.springernature.com/orcid.

We hope to receive your revised paper within three weeks. If you cannot send it within this time, please let us know.

Yours sincerely,

Reviewers Comments:

Reviewer #1 (Remarks to the Author):

The revised manuscript by Steindorff and colleagues is much improved, with my original comments addressed satisfactorily. However, some sections and figures could use additional editing to increase clarity and impact.

Line 75 I would say "Results identify Trichoderma as an ancient, genetically cohesive..."

Line 133. Please define "HOGs".

Lines 149-164. The section "The Start of Trichoderma Life Cycle..." needs an introductory sentence explaining the rationale for the presented experiments and why the phyllosphere microbial mats are important in the first place. The paragraphs seem choppy, disconnected from each other and devoid of a broader synthesis.

Line 170. Please explain the term "idioadaptation".

Line 204. I do not follow the opening comment "To further assess Trichoderma fitness..." Do the authors mean "fitness-related traits"?

Lines 300-301. I would say "Ecological Adaptations in Trichoderma Genomes Mediated by Regulatory Genes, Small Secreted Proteins, and Genes of Unknown Function."

Line 302. I would say "We demonstrated that Trichoderma spp. maintain genomic cohesion while undergoing adaptive radiation and developing significant ecophysiological variability..."

Figure 2. I found panel (a) not very informative and would suggest showing only one species per strategy. The combination of panel (b) and panel (c) is great. However, I do not see the colors mentioned in the caption to be represented in panel (c). Please adjust the diagram in panel (c) to represent these colors.

Figure 3. I love the info content of this figure. However, I would suggest using more colors and shades to denote responses on different substrates in (a). Also, using shades of purple and green for both taxonomy and responses is confusing.

Supplementary Figure 4, which provides support for most of the section "Profound Edaphic Incompetence...", conveys a massive amount of information. Please provide a more detailed and meaningful caption describing this content.

Reviewer #2 (Remarks to the Author):

All my concerns have been satisfactorily addressed

Reviewer #3 (Remarks to the Author):

I thank the authors for addressing my comments, particularly regarding the accessibility of the code used in to process the data presented in the manuscript. I still have some remarks regarding a few points in the revised manuscript.

I appreciate that the authors have now referenced Fig. 3b and clarified its intended contrast with phenotypic variability in Fig. 3a. However, the integration remains largely descriptive, and the term "genome inventories" overgeneralizes what is shown (specific gene category counts rather than total genomic content). To justify inclusion of panel b in the main figure, the authors should provide a minimal analytical or quantitative link, such as a measure of variance in these genomic features, a comparison of their range relative to phenotypic variation or a statement on whether any of these categories correlate with fitness metrics. If such analysis is beyond the present scope, I recommend moving panel b to the Extended Data or omitting it altogether, since its current treatment does not meaningfully advance the results or interpretation.

As a minor note, I noticed some inconsistency between the original and revised Figure 1:

T. inhamatum, *T. spirale* and *Clonostachys* have different substrates between the two figures. In the revised version, *T. inhamatum* does not have "soil saprotrophy", *T. spirale* has the symbol for 'endophyte, epiphyte, parasite, rhizosphere colonizer', which was absent in the first version, and *Clonostachys* has two new symbols for 'mycoparasite' and 'litter decomposition'.

Reviewer #4 (Remarks to the Author):

This manuscript provides a valuable and highly ambitious study, integrating comparative genomics with large-scale ecophysiological, reproductive, stress-tolerance, and biotic-interaction assays across 37 strains. It has the potential to become a landmark reference on *Trichoderma* biology. The study also highlights biosafety concerns regarding opportunistic species, such as those with human or plant pathogenicity, especially in the context of their use as biocontrol agents or biofungicides in agriculture. I congratulate the authors for the extensive analyses conducted and the formidable dataset generated and presented. The authors have now streamlined the narrative, compared to an earlier version, and provided significant clarifications in response to earlier critiques. There is still some overinterpretation in some sections that could be tempered down.

Major comments:

The authors state that contrasting phenotypic profiles among sympatric or closely related taxa indicate character displacement. However, true character displacement would require evidence of niche partitioning driven by interspecific competition within sympatry. Here, sympatry is assumed but not demonstrated (e.g., no biogeographic co-occurrence data). Differences in BIOLOG phenotypes alone are insufficient. Perhaps the claims could be rephrased as "consistent with" or "suggestive of" character displacement.

The authors highlight *T. afroharzianum* and *T. longibrachiatum* as high-concern species from a biosafety perspective, referencing plant and human pathogenicity. However, strain-level variation could be important and should probably be considered instead of making species-level generalizations. Agronomic recommendations rely on regulatory guidance that would focus on a strain that would be submitted to the regulatory process. Species are not regulated, but strains are. I think that the biosafety section is important and provides a solid application of the findings of this work, but it would be prudent to emphasize that the data could constitute a first-tier screening, not prescriptive regulatory conclusions.

The authors addressed the rest of the comments appropriately in the manuscript and the rebuttal letter.

Version 2:

Decision Letter:

Our ref: NMICROBIOL-25041300B

8th December 2025

Dear Dr. Druzhinina,

Thank you for submitting your revised manuscript "Phenogenomic determinants of ecological plasticity and fitness in the genus *Trichoderma*" (NMICROBIOL-25041300B). It has now been seen by the original referees and their comments are below. The reviewers find that the paper has improved in revision, and therefore we'll be happy in principle to publish it in *Nature Microbiology*, pending minor revisions to satisfy the referees' final requests and to comply with our editorial and formatting guidelines.

Thank you again for your interest in Nature Microbiology. Please do not hesitate to contact me if you have any questions.

Sincerely,

Version 3:

Decision Letter:

8th January 2026

Dear Professor Druzhinina,

I am pleased to accept your Resource "Phenogenomics reveals the ecology and evolution of Trichoderma fungi for sustainable agriculture" for publication in Nature Microbiology. Thank you for having chosen to submit your work to us and many congratulations.

After the grant of rights is completed, you will receive a link to your electronic proof via email with a request to make any corrections within 48 hours. If, when you receive your proof, you cannot meet this deadline, please inform us at rjsproduction@springernature.com immediately. You will not receive your proofs until the publishing agreement has been received through our system.

Authors may need to take specific actions to achieve compliance with funder and institutional open access mandates. If your research is supported by a funder that requires immediate open access (e.g. according to [Plan S principles](https://www.springernature.com/gp/open-science/plan-s-compliance) or the [NIH public access policy](https://www.springernature.com/gp/open-science/us-federal-agency-compliance)) then you should select the gold OA route, and we will direct you to the compliant route where possible. Because authors warrant under our subscription licensing terms that they haven't committed to licensing any version of their article under a licence inconsistent with the terms of our agreement – including the applicable embargo period – publication under the subscription model isn't suitable for authors whose funders require no embargo.

We welcome the submission of potential cover material (including a short caption of around 40 words) related to your manuscript; suggestions should be sent to Nature Microbiology as electronic files (the image should be 300 dpi at 210 x 297 mm).

in either TIFF or JPEG format). Please note that such pictures should be selected more for their aesthetic appeal than for their scientific content, and that colour images work better than black and white or grayscale images. Please do not try to design a cover with the Nature Microbiology logo etc., and please do not submit composites of images related to your work. I am sure you will understand that we cannot make any promise as to whether any of your suggestions might be selected for the cover of the journal.

With kind regards,

P.S. Click on the following link if you would like to recommend Nature Microbiology to your librarian
<http://www.nature.com/subscriptions/recommend.html#forms>

** Visit the Springer Nature Editorial and Publishing website at http://editorial-jobs.springernature.com?utm_source=ejP_NMicro_email&utm_medium=ejP_NMicro_email&utm_campaign=ejp_NMicro for more information about our career opportunities. If you have any questions please click [here](mailto:editorial.publishing.jobs@springernature.com).

Open Access This Peer Review File is licensed under a Creative Commons Attribution 4.0 International License, which permits use, sharing, adaptation, distribution and reproduction in any medium or format, as long as you give appropriate credit to the original author(s) and the source, provide a link to the Creative Commons license, and indicate if changes were made. In cases where reviewers are anonymous, credit should be given to 'Anonymous Referee' and the source. The images or other third party material in this Peer Review File are included in the article's Creative Commons license, unless indicated otherwise in a credit line to the material. If material is not included in the article's Creative Commons license and your intended use is not permitted by statutory regulation or exceeds the permitted use, you will need to obtain permission directly from the copyright holder.

Editor's comments

In particular, we suggest that you make major changes to the manuscript to address the technical concerns with the computational methods, improve the claims made regarding the implication of this work in sustainable agriculture, address the conflicts highlighted by the referees with the current literature, improve the code/data availability, and provide more context with improved discussion. Should this allow you to address these criticisms, we would be happy to look at a revised manuscript.

Response:

We greatly appreciate the editor's guidance and the referees' constructive comments. In the revised manuscript we have addressed all of these points. The computational methods are now fully clarified and documented, with code and reproducibility packages deposited in a public repository; claims regarding sustainable agriculture and biosafety have been refined and grounded in a dedicated framework (Extended Data Fig. 6). Ecological concepts have been cross-checked against the literature and terminology refined for precision. Finally, the entire manuscript has been streamlined and tightened to meet the journal's length requirements.

Referee #1: Fungal evolutionary biology (Remarks to the Author):

The manuscript by Steindorff and colleagues reports on a survey of genomic and phenotypic data across 34 species representing the genus *Trichoderma*. This formidable effort to link genomic and phenotypic variation across a range of closely related species of economically important fungi is highly commendable. However, the language used to describe the proposed conceptual framework is confusing and needs to be refined.

Response:

We sincerely appreciate the thoughtful critique. We have clarified the conceptual framework and terminology, added missing citations and numeric ranges, tightened speculative language, and improved figure legibility.

In particular, the authors seem to introduce new terms "phenogenes", which they define as genes linked to fitness-related phenotypes (line 83) and "phenotype-associated homologous orthologous groups, (phenoHOGs)" that are statistically significantly correlated with distinct phenotypic traits (line 373). I am baffled by these definitions and the casual way they are presented. I am familiar with the term "orthologous phenotypes", or "phenologs", which predict unique genes associated with specific phenotypes and reveal functionally coherent, evolutionarily conserved gene networks (McGary 2010, www.pnas.org/cgi/doi/10.1073/pnas.0910200107). From the presented narrative, I gather that the authors attempt to introduce their new terms to identify sets of genes that underlie phenologs. If so, this effort requires further development.

Response:

We agree that terminology must be precise and unambiguous. Phenologs, sensu McGary et al. (2010)¹, are defined as phenotype–phenotype relationships across species based on statistically significant overlap of orthologous genes. By contrast, our study addresses genotype–phenotype associations within a clade of closely related species. In the original manuscript, we used the term

***phenogenes* to describe genes linked to measured phenotypes, but we concur that introducing a new term is unnecessary. In revision, we now use the straightforward descriptor *phenotype-associated orthogroups (PAOGs)*—orthogroups whose presence/absence pattern across the *Trichoderma* panel is significantly associated with a phenotype—and explicitly clarify how this differs from phenologs. This terminology does not propose a new biological category but simply denotes orthogroups with significant phenotype associations in our dataset. We also added a clarifying statement in the Results to distinguish PAOGs from phenologs.**

I also have a problem with comments on character displacement, which do not seem to have any geographic context, as no data on geographic distribution of focal species are presented (lines 84-86 and 318-327).

Response:

We highly appreciate this comment and agree that the geographic context is crucial for the discussion of putative character displacement. The origins of the strains are provided in Supplementary Table 1, and the dataset was designed to include both sympatric and allopatric strains. We have revised the text to (i) clarify that our data show pronounced phenotypic divergence among closely related strains isolated from overlapping habitats (e.g. arboreal epiphytes in Southeast Asia; Supplementary Table 1), and (ii) reference published cases of sympatric species in *Trichoderma*. Sympatric occurrence of closely related *Trichoderma* species is in fact a common phenomenon, well documented in the literature—for example, between *T. longibrachiatum* and *T. orientale*², *T. asperellum* and *T. asperelloides*³, within the *T. harzianum* complex⁴, and most strikingly between *T. harzianum* and *T. guizhouense*, where experimental evidence demonstrated disruptive selection on hydrophobin genes and contrasting dispersal strategies despite co-occurrence⁵. We therefore interpret the phenotypic divergence we observe as consistent with character displacement, while explicitly noting that formal demonstration would require rigorous sympatry mapping. To reflect this more cautious interpretation, we have also removed the phrase “Drastic Character Displacement” from the section title and replaced it with a more neutral description of phenotypic divergence.

Lesser problems:

Line 113. Please include missing citations.

Response:

Done.

Line 158. Please state the range of gene sizes in addition to referring to Extended Data Figure 1.

Response:

Done — we have specified the range of genome sizes and gene counts (Extended Data Fig. 1)

Lines 170-175. Where in Fig 1 is this information about results of the gene coevolution network analysis displayed?

Response:

The results are shown in the six pie charts on the right-hand side of Fig. 1, which summarize the composition of taxon-specific orthogroups (PFAM-

annotated proteins, SSPs, and genes of unknown function), the size of the taxon-specific genome (numbers above each pie), and the proportion of co-evolving genes (percentages below). We have revised the legend to make explicit that these pie charts represent the outcomes of the gene coevolution network analysis.

Lines 176-181. I appreciate this conclusion. However, I wonder whether it is fully justified, given that it is based on only 34 out of 485 species validly described in the *Trichoderma* genus (line 103).

Response:

Our dataset was designed to capture the full scope of infrageneric diversity in *Trichoderma*, with representatives from nearly all major clades. Although only 34 species were profiled, they were selected as phylogenetically and ecophysiological informative exemplars, ensuring that the principal evolutionary lineages are represented⁶. This study represents the first phase of a whole-genus genomic initiative that will expand coverage further. Given the recent intensive taxonomic splitting in *Trichoderma*, which has produced many validly described but biologically similar species, we are confident that our design provides robust support for the conclusions.

Line 197-201. These conclusions seem highly speculative. I would leave them for the Discussion section.

Response:

We have moved this interpretive statement from the Results to the Discussion.

Line 241-245. Please include a citation for info on sialic acids.

Response:

Thank you, the reference⁷ is now included:

Line 338-340. Please specify enzymatic abilities underpinning these traits.

Response:

We clarified that this section synthesizes results from qualitative assays (agar-plate proteolysis, dual-confrontation mycoparasitism, hydroponic plant-growth). These assays capture secretome-/metabolome-level functional outputs (e.g., casein clearing; overgrowth/lysis; inhibition/stimulation) and are therefore summarized as “proteolytic potential” and “mycoparasitic vigour” (see Methods and Supplementary Fig. 4).

Line 408. What is a “postgenomic toolbox”?

Response:

We now use “integrating comparative genomics with ecophysiological profiling” to avoid ambiguity.

Line 428. Please correct a typo in “expansins”.

Response:

Done.

Fig 1. Cartoon lifestyle identifiers are too complex and difficult to distinguish. Please simplify. It is difficult to distinguish the lifestyle legend from the phylogeny. Please differentiate.

Response:

Thank you for this suggestion. We have simplified the cartoons, separated the legend from the phylogeny, and streamlined the figure to improve clarity.

Fig 3a. “Loinear”?

Response:

Done.

Fig 3b. Please explain abbreviations in the caption.

Response:

Done.

Extended Data Figure 2. Please explain in the caption the meaning of “germination profile” and the scale from -2 to 8.

Response:

We added the explanations and the scale. Thank you.

Referee #2: *Trichoderma* (Remarks to the Author):

Introduction:

Biotrophic Trichoderma strains can also directly control plant pathogenic fungi and nematodes, positively affecting crop microbiomes- Needs some clarification here.

The genus is primarily saprophyte, sometimes behaving as hemibiotrophs while attacking other fungi. Here the authors claim that some are biotrophs. Please discuss this aspect elaborately, will be good for readers to get a clear idea about their existence in nature- as saprophyte, hemibiotroph or biotroph, or all of these!

Response:

We agree that terminology must be precise and have refined the Introduction accordingly, without increasing its length. We now avoid the generic term “biotrophic” and instead specify the interaction modes relevant to *Trichoderma*: mycoparasitism, facultative plant endophytism, occasional plant pathogenicity, and opportunistic infections in immunocompromised humans (facultative animal biotrophy). Comparative genomics and biodiversity surveys indicate that mycotrophy is pervasive and likely ancestral in *Trichoderma*^{8,9}, whereas efficient phytosaprotrophy appears to be a derived expansion linked to acquisition and diversification of plant cell wall-degrading CAZymes¹⁰. This clarification preserves the original logic while aligning the framing with genus-wide evidence, making clear that *Trichoderma* species in nature can act as saprotrophs, hemibiotrophs, and biotrophs.

Several species can also become endophytic in roots, stems and leaves of wooden plants(refs)- please include references, seems like an unedited version!

Response:

Included—thank you for catching this.

Additionally, reports of crop pathogenicity of Trichoderma become common- or should it be becoming common?

Response:

Rephrased as suggested. Thank you.

Results:

The optimal carbon source for all Trichoderma strains was N-acetyl-D-glucosamine, a monomer of chitin found in fungal cell walls and the exoskeletons of arthropods. This preference underscores mycoparasitic capabilities of Trichoderma, its competitive interactions within fungal communities and occasional parasitism on insects- many Trichoderma are aggressive parasite on oomycetes (lacking chitin in their cell wall), please through some light on this aspect.

Response:

We agree that antagonism of cellulose-walled oomycetes is common in *Trichoderma* and is chitin-independent. Our N-acetyl-D-glucosamine statement is strictly tied to the carbon-source assay in this section and suggest a link to pervasive mycotrophy; it does not address oomycete interactions. In a companion genus-wide analysis of the same strain panel, we showed that antagonism of *Trichoderma* to *Pythium/Globisporangium* is widespread, with cellulases contributing among multiple factors¹¹. We now note it in the Discussion.

Limited growth on a number of specialized substrates reveals ecological conditions likely unfavorable for Trichoderma colonization, possibly characterized by anaerobic environments- if I understood correctly, the authors are emphasizing that phyllosphere is the main habitat of Trichoderma. Phyllosphere does not really represent an anaerobic environment, or does it? This point needs to be discussed for better clarity.

Response:

Thank you for this request. We have clarified the text to avoid confusion with the phyllosphere and to specify that the limited growth reflects stress on certain substrates:

“By contrast, limited growth on specialized substrates—particularly organic acids (fumaric, succinic) and amino acids (L-glutamic, L-asparagine)—indicates metabolic stress and points to environments such as acidic or fermentative conditions that are unfavorable for *Trichoderma* colonization (Extended Data Fig. 3).”.

Genotypic Diversity and Nutritional Opportunities Influence Reproductive Strategies of Trichoderma strains: No discussion on chlamydospores.

Response:

We appreciate this valuable point and fully agree that chlamydospores are important reproductive structures in *Trichoderma*. However, their formation is highly condition-dependent and not comparable across strains under a single standardized regime¹²; absence under our conditions would thus reflect suboptimal induction rather than true inability. For the same reason, we did not include mating potential in the genus-wide panel, even though many species form sexual structures in nature¹³. Despite multiple attempts under different conditions and compatible mating types, in vitro mating was only observed in *T. reesei*, making cross-strain comparisons uninterpretable. To ensure reliable comparisons across the entire dataset, we therefore focused our reproductive trait set on conidiation, which provided consistent and quantitative outputs. Space constraints preclude a fuller discussion here; we intend to address this aspect in subsequent work.

Negligible Antibacterial Activity: Not really universal, some metabolites, like gliotoxin, exhibit strong antimicrobial activity.

Response:

Thank you. This statement in the results refers to the standardized, genus-wide plate assays used here; under these conditions, antibacterial activity was uniformly low across the panel, however there were some positive cases. We now note in the Discussion that lineage-specific primary antifungal secondary metabolites can also show antibacterial effects—e.g., gliotoxin from some *T. virens* and peptaibols from several species—although these activities are not universal^{14,15}.

Discussion:

..perfectly adapted for mycoparasitism within phyllosphere-associated microbial biofilms (bark, leaves, roots)..- this is very interesting and novel insight. However, do we consider roots as part of phyllosphere?

Response:

Thank you, we corrected the wording.

..robust but non-circadian response to light..- Are the authors contradicting such studies: "Circadian oscillations in *Trichoderma atroviride* and the role of core clock components in secondary metabolism, development, and mycoparasitism against the phytopathogen *Botrytis cinerea*. Henríquez-Urrutia M, Spanner R, Olivares-Yáñez C, Seguel-Avello A, Pérez-Lara R, Guillén-Alonso H, Winkler R, Herrera-Estrella A, Canessa P, Larrondo LF. *Elife*. 2022 Aug 11;11:e71358. doi: 10.7554/eLife.71358", if so please indicate clearly.

Response:

Thank you for raising this. We do not dispute the existence of a functional circadian clock in *T. atroviride*¹⁶. Our statement was intended at the genus scale: across the broader *Trichoderma* panel profiled here, we quantified robust light-induced developmental responses but most did not resolve circadian rhythmicity. We have revised the wording to avoid over-generalization.

Unfortunately, testing the hypothesis that Trichoderma is predominantly associated with tropical rainforest canopies and becomes edaphic in temperate or polar regions is currently unfeasible due to a sampling bias,...- towards the end of the manuscript, this is a confusing statement (at least I'm confused), with previous discussion on phyllosphere as the primary habitat; this can be put in a better way.

Response:

We are confident the revised version has no contradictions. Thank you for addressing it.

Additionally, the proposed affinity to the phyllosphere points to biotrophic associations with plants- is this true? Are they really biotrophs with no saprophytic existence?

Response:

We specified that we mean facultative biotrophic associations with plants. We appreciate the reviewer's caution and have softened the statement.

General:

Well defined strain variability, such as those existing in *T. virens*, not accounted for.

Response:

We agree that intraspecific variability—well documented in *T. virens*, *T. harzianum*, *T. asperellum*, *T. longibrachiatum*, and others—is important, particularly given the shallow or unresolved species boundaries within the genus. Our dataset was designed to capture variability both within and between species (e.g., sympatric and closely related pairs), and we observed pronounced strain-level divergence across multiple readouts (relative fitness map, Fig. 3a; nutritional “barcodes,” Extended Data Fig. 4; REPAINT/developmental traits, Supplementary Data). These results directly support our central claim of ecological plasticity and niche-linked trade-offs. To accommodate such heterogeneity without unduly expanding the manuscript, we emphasized phenotype-associated orthogroups (PAOGs) rather than taxonomic rank, providing a scalable gene-centric framework that is less constrained by unresolved species concepts. We agree that extending this approach to populations of cosmopolitan and common species such as *T. virens* or *T. longibrachiatum* would be a valuable direction for future studies.

Secondary metabolism, especially presence of known mycotoxins (a significant risk to safe use), has not been studied/discussed.

Response:

We appreciate this important point. Indeed, a comprehensive genus-wide assessment of secondary metabolism in *Trichoderma* is currently beyond reach, as our understanding of the production, regulation, and genetic control of most compounds is still fragmented. Only a few metabolites, such as peptaibols and 6-pentyl- α -pyrone (6-PP), are known to occur across large parts of the genus, and even these require further work to compare their diversity and conditions of induction. For the majority of secondary metabolite families, the available evidence indicates strong taxonomic restriction (e.g., gliotoxin and trichodermamides in *T. virens*¹⁴; viridins in the *T. viride* complex; koninginins in the *T. koningii* group; harzianolide and harzianopyridone in the *T. harzianum* clade^{17,18}; trichothecenes in *T. brevicompactum*¹⁹ and *T. arundinaceum*; sorbicillins in the *Longibrachiatum* clade²⁰). While these metabolites are highly relevant for biosafety considerations, their restricted phylogenetic distribution means that they do not allow genus-wide comparison within the framework of the present study. We fully agree that this is a priority area for future research, and our discussion now clarifies why a systematic evaluation of *Trichoderma* secondary metabolism was not included here.

Phenotypes that have been published already should have been taken into account. The authors' aim was to "provide an evolutionary framework for understanding fitness trade-offs between environmental opportunism and ecological specialization". However, discussion on why some species/strains are human pathogens, a very important aspect related to safety, is lacking. Similarly, can we predict if a particular strain has the potential to be a plant pathogen? This is very important as it's not possible to assess the behaviour on all plants before commercial release of a formulation.

Response:

We also appreciate this comment. We note that the extensive literature on *Trichoderma* is strongly skewed toward particular traits depending on the model species—for example, cellulase activity is usually assessed in *T. reesei*, mycoparasitism in biocontrol strains, and light and injury response in *T. atroviride*. To avoid reinforcing these biases, our study was designed to

provide a genus-wide, standardized, and comparative phenotyping framework that treats all species equally across a broad range of traits, allowing us to detect convergent and divergent patterns without being limited by historical focus.

With respect to pathogenic potential, our dataset cannot directly predict whether an individual strain will cause disease in plants or humans, as these outcomes depend more on opportunistic vigour and host immunity than on *Trichoderma* specialization. However, we identify trait and gene combinations correlated with opportunism—such as stress-resistant spores, aerial dispersal, bulk-soil growth, and nutritional versatility. These features provide risk indicators that we explicitly incorporate into our proposed first-tier biosecurity framework (Extended Data Fig. 6). We therefore view our work as a step toward systematic risk assessment, while recognizing that definitive predictions of pathogenicity will require integration with host-specific assays.

The discussion begins with "The aim of our study was to utilize a postgenomic toolbox to provide understanding of *Trichoderma* ecology and evolution, which underpin the science-based risk assessment of its application in sustainable agriculture", however, at the end of the day, do we have such a toolbox using which we can assess the risks?

Response:

We now use “integrating comparative genomics with ecophysiological profiling” to avoid ambiguity. See also our reply to Reviewer 1.

Some *Trichoderma* (like *T. virens*) spores are embedded in gelatinous matrix and hence not air-borne, while most others are produce dry spores which are easily air-borne. Did the authors look for such phenotype in the strains under study?

Response:

We thank the reviewer for raising this point. Indeed, spore morphology and embedding in a gelatinous matrix, as reported for *T. virens* and for other species, i.e. occasionally for *T. harzianum* sensu stricto, are important for dispersal ecology. To capture these differences in our dataset, we explicitly tested both pluviophilous dispersal (water droplet-mediated) and anemophilous dispersal (air-borne) across all strains (see Methods, Fig. 3a, Supplementary Figure 3). Thus, the variation noted by the reviewer is represented in our assays and is reflected in the comparative dispersal profiles reported.

Referee #3: Machine learning and genomics (Remarks to the Author):

This manuscript offers a comprehensive insight into *Trichoderma* and overall the work is of high quality, with good and clear **writing and was very pleasant to read.**

Response:

We thank the reviewer for the generous and encouraging assessment.

I have a selection of major and minor comments, which could help improve further this manuscript. But first, I would like to address a critical concern regarding data and algorithm availability. In the corresponding sections (data availability and code availability), the authors

wrote that ‘additional data supporting the findings are available from the corresponding author upon reasonable request’ (line 519-521) and ‘The REPAINT AI algorithm (...) is available from the corresponding author upon reasonable request’ (line 523/534). The withholding of data and algorithm without clear and adequate reasons is not acceptable and not compatible with high quality work put together in this manuscript. Gatekeeping practices do not offer any benefits to the community and can only lead to loss of data and reproducibility in the long term, as highlighted in this paper: doi.org/10.1038/s41597-021-00981-0.

Response:

We thank the reviewer for raising this important point regarding transparency and long-term reproducibility. We would like to emphasize that we are fully committed to open science and to making all relevant data and code available.

For the co-evolution analyses and SVM predictions, we have prepared a complete package including the code and processed data files, each accompanied by a detailed README. These have already been shared in a <https://code.jgi.doe.gov/assteindorff/jgi-trichoderma> public repository, which was added to the methods section. The underlying genomic datasets are already freely accessible through the JGI MycoCosm portal (<https://mycocosm.jgi.doe.gov/trichocosm/trichocosm.info.html>).

Regarding REPAINT, the algorithm was developed by our co-authors Thomas Ebner, Philipp Kainz, and Michael Mayrhofer-Reinhartshuber at KOLAIDO GmbH (formerly KML Vision GmbH) under a research contract and was described in detail in Cai *et al.* 2020⁵. The software is maintained and distributed by the company and is available via their website (<https://kmlvision.atlassian.net/wiki/spaces/KB/pages/3450011692/Fungi+REPAINT+App+Versions>), where instructions for use are provided.

List of major comments:

Result ‘Ecological Adaptations in Trichoderma Genomes Enhanced by Regulatory Genes, Small Secreted Proteins, and Genes of Unknown Function’. This section is rather confusing to read (see comments later about the associated methods section). I cannot find the “more than 150 parameters” in the Supplementary Data 2. Or does it correspond to Supplementary Data 2D? In which case the number of parameters is 141?

Response:

We are sorry for the inconsistency. The original statement was an earlier miscalculation. In the final version deposited as Supplementary Data 2D, the number of parameters is 141. We have corrected the text accordingly to ensure consistency with the Supplementary Data.

Does the “uncertainty of each trait” (line 372) means that the authors were not able to test for the traits, or that the results were not conclusive?

Response:

The “uncertainty” designation does not mean that traits were untested or non-reproducible, but that the response was heterogeneous and did not allow for an unambiguous binary call. For example, in mechanical injury assays, some strains produced abundant conidia directly at the scar, others conidiated only around it, while some showed only scattered conidia in the

scar area. Such intermediate or irregular outcomes were marked as “uncertain.” We have clarified this in the text.

Figure 2: the regression lines don't add any value to the figure, except showing that they are a poor fit. And they are not used in the main text.

Response:

We agree with the reviewer and have removed the regression lines from Fig. 2.

The PCA plot needs to have clear axes (Principal component 1, principal component 2) and the amount of variation they explain.

The coloring in the PCA is also confusing with some strain having two colors, please clarify why.

Response:

Done. Axes are now labelled with explained variance, and each strain is shown with a single, consistent colour.

Overall, the use of average growth rate across all carbon sources is misleading and hides the variation of growth rate per strain across multiple carbon sources.

Response:

The strain- and carbon-specific growth rates are fully presented in Extended Data Fig. 3 and Supplementary Figs. 1–2. In Fig. 2, bubble size represents the mean growth rate across all carbon sources, used only as a synthetic summary measure to highlight overall trends across strains. We have revised the legend of Fig. 2 to clarify this.

Figure 3b. I don't see where that figure is used/mentioned in the text. Figure 3 is referenced at line 291, 296, 314, 330, 333 but only for information regarding Fig 3a. Make sure Figure 3b is referenced and used in the main text, or consider removing it from the Figure. Also, the acronyms (GT, GH, CE, PL, SM, TC, NRPS, PKS) used in Fig 3b are not explained in the legend (or in the text).

Response:

To improve the role of Fig. 3b, we now explain explicitly how it complements panel a:

“While Fig. 3a highlights variability in fitness traits, Fig. 3b shows that the gene counts (number of proteins, transcription factors, transporters, major groups of enzymes, biosynthetic gene clusters) remain comparatively compact. This asymmetry highlights that phenotypic differences among strains are more pronounced than variation in overall genome inventories (see Extended Data Fig. 1 for details).”

We also updated the figure legend to define all acronyms.

Figure 5: Panel C and D are not used or referenced in the main text (panel D corresponds to the last sentence in the result section). Consider removing them, or discuss their relevance in the main text. Panel C is hardly readable, both from the multiple lines linking phenotypes, to the phenotype colors (circle line too thin). An alternative could be a heatmap-like figure. In panel B, some transcription factors are found only in phenogenes (asterix mark) but not in the genomes? It raises the question of their origin. Or did the author compute the TF found

in phenogenes, and the ones found in the REST of the genomes? Please clarify this in the figure and legend.

Response:

We thank the reviewer for these constructive suggestions. Panel c has been removed, while Panel d has been retained as it provides critical evidence for the reduced number of PAOGs in the *Longibrachiatum* clade. We have revised both the Results and Discussion to explicitly reference and interpret this panel, and we have clarified its description in the legend. In Fig. 5b, the comparison contrasts TF families found among PAOGs with those in the remainder of the genome. Families marked with an asterisk are not novel TFs absent from the genome, but categories recovered exclusively from the PAOG subset in this analysis. This clarification has been added to the legend of Fig. 5 and the legend of the corresponding Extended Date Figure.

Methods ‘Genome sequencing an assembly’.

Line 1024-1025 “An automated attempt was made to reassemble any potential organelle (mitochondrion) from the filtered reads and remove any organelle-matching reads with kmer matching against the resulting contigs with an in-house tool.”. Code should be available for full reproducibility.

Response:

Done. The organelle-matching reads were filtered using `bbduk.sh [-Xmx110g k=25 mm=f mkf=0.03 ordered ow]`, part of the `bbtools` package (<https://bbmap.org/>). This sentence was included in the ‘Genome sequencing and assembly section’.

Methods “genomic feature selection”. Important information is missing: it is not clear what is the training dataset and the tested dataset (genomes and phenotypes), what kind of kernel was used (linear?) and how the correlation values were obtained (Pearson?).

Response:

The input data is gene counts built with all *Trichoderma* amino acid sequences. Each row is an orthologous group (OG) and each column a genome. For training/test we used the genomes for which the phenotype was defined (“Phenomics” section). A jack-knife leave-one-out approach was used to generate the *accuracy* score for each feature combination. These scores were used, not correlation. We removed it from the text.

Please add the version of scikit-learn used, as “`sklearn.cross_validation`” is not part of scikit-learn since 2016 (v0.18). Overall, it would be better to provide the full code (github repository, zenodo, figshare, etc) for clarity and reproducibility of the results.

Response:

We provided the complete code on <https://code.igi.doe.gov/assteindorff/igi-trichoderma>, including the `environment.yml` file to build the Conda environment. It contains the versions of all packages used. The repository contains all the code and files used in the manuscript.

List of minor comments:

Line 113: reference missing (noted “(refs)”)

Response:

Done, thank you.

Line 123-124, "...the identification of phenogenes - genes enhancing phenetic traits": as the authors define phenogenes later (line 130), these genes are associated with phenotypes, rather than 'enhancing' them.

Response:

We thank the reviewer for noting this imprecision. In the revised manuscript we no longer use the term "enhancing," but instead state that these orthogroups are *associated with* phenotypes. This change aligns with our broader clarification (also see response to Reviewer 1) where we replaced "phenogenes" with the straightforward term *phenotype-associated orthogroups (PAOGs)* to avoid ambiguity.

Figure 3a. The last column "product" is not defined, but I am assuming it is the product of the three fitness related-trait. It would be nice if clarified in the legend.

Response:

We have clarified in the legend that the "Product" column represents the multiplicative composite of the three fitness-related traits (development, stress tolerance, and dispersal).

Similarly, I am assuming that the stars represent cold and drought resistant strains, but their annotation is not consistent (lacking for the first two stars) and could be described in the legend.

Response:

We thank the reviewer for catching this. The stars indicate strains whose spores did not survive cold or drought, resulting in a product value of zero. This has now been specified in the figure, and the annotation corrected for consistency.

Figure 4a: Why is the phenotype associated represented in a two dimensional plot when it is a one dimension value (prob of 1 = 1 - prob of 0). Clarity could be improved with a simple vertical dot plot or something similar.

Response:

For Figure 4a, we considered each phenotype's presence or absence as a dimension to better visualize the separation between the predictions. We find it an easy way to visualize phenotypes where the orthologous groups are capable of distinguishing the species by phenotype (closer to 0-1 or 1-0) or not (surrounding 0.5-0.5 region of the plot).

Methods: missing software version for AlphaFold2, Foldseek, PhyKIT, scikit-learn.

Response:

Done. All versions are included in the manuscript.

Referee #4: Fungal genomics (Remarks to the Author):

This is an ambitious and comprehensive study that combines genomics, phenotyping, and ecological profiling of *Trichoderma* spp. to understand their ecological versatility and to help

assess their implications in terms of agricultural applications and also the risk that they pose to crops and animals.

The study is based on the premise that while *Trichoderma* species hold enormous promise as biofungicides and growth-promoting agents, and as they are increasingly deployed across fields and crops, their evolutionary plasticity and occasional pathogenicity require careful risk assessment.

This is a strong premise and makes a strong argument for the proposed work. And the work reported make a valuable contribution to fungal ecology and evolutionary biology; but the manuscript in its current feels unfocussed and the main core of the paper is lost in the abundance of the data.

Response:

We thank the referee for recognizing the ambition and scope of this ecological-genomics study. Despite the ubiquity of *Trichoderma*, its biology has until now been interpreted largely through the prism of application, with attention focused on a few agronomically or industrially relevant strains. Earlier genomic studies^{9,8}, necessarily restricted to such models, revealed substantial genetic divergence but a surprisingly narrow and convergent set of crop-associated traits. At the same time, the use of *Trichoderma* in sustainable agriculture is expanding rapidly²¹, and reports of adverse effects on mushrooms, humans, and occasionally plants are becoming more frequent. This dual trend underscores the need to move beyond an application-driven perspective and to establish a genus-wide framework that connects evolutionary breadth with ecological function.

By sampling across the phylogenetic diversity of this ancient lineage and assaying a broad spectrum of ecologically informative traits, we demonstrate phenotypic variation that in many cases exceeds genomic divergence. This approach provides a more balanced view of genotype–phenotype linkages, independent of a few model strains, and offers fresh insight into the broader biology of these fungi. While it does not specifically resolve every open question, it establishes a framework for future studies: highlighting candidate genes most likely tied to ecophysiological differentiation, identifying phenotypes that reveal ecological speciation, and showing that apparent “cryptic species” can be distinguished when ecological and genomic data are integrated.

We also acknowledge the referee’s concern that the breadth of data risked obscuring the central message. In revision, we have considerably streamlined the Results and Discussion, reduced redundancy, and sharpened the narrative, while retaining the comprehensive scope required to capture the ecological versatility of the genus.

Fig 3 contains a donating amount of information and it is not clear whether or not all of it is required.

Response:

Fig. 3 is intentionally information-dense, as it juxtaposes broad phenotypic divergence (Fig. 3a) with relatively compact genomic inventories (Fig. 3b). To improve clarity, we streamlined the legend and the corresponding Results text, while keeping Extended Data Fig. 1 available for readers seeking full genomic detail.

The manuscript is dense and at times overloaded with detail, which can obscure the main conclusions. In particular, the link between the work reported and the biosafety risk is weak and almost appears as an afterthought, after the first paragraph of the introduction.

None of the figures, it seems, contribute to that question of biosafety risk, or at least, not clearly. A more streamlined version, possibly emphasizing ecological and evolutionary implications, with a tighter focus and more concise presentation would improve clarity and make for a more impactful contribution. In particular, if the focus is to be on balancing biosafety risks, as introduced at the beginning of the paper, via an evolutionary framework, there needs to be a better narrative. Alternatively, the authors could just drop the biosafety risk claims and just focus on the ecology and evolution of the group.

Response:

We thank the referee for emphasizing the need to articulate the biosafety dimension more clearly. In response, we revised both the Results and Discussion and added a dedicated framework (Extended Data Fig. 6) that links the phenotypes measured here (e.g., dispersal, spore resilience) with literature reports of human and plant pathogenicity. This framework is explicitly non-prescriptive: it does not assign hazard but organizes evidence into a first-tier screen to support context-specific evaluation and safeguards.

We also strengthened the narrative by integrating biosafety considerations throughout the manuscript rather than confining them to the Introduction. The agronomic use of *Trichoderma* is relatively recent—just over fifty years—and long-term field data are lacking. While its application was initially modest, *Trichoderma*-based products have expanded rapidly in recent years, with market analyses predicting continued growth. Against this backdrop, reports of adverse plant effects are becoming more frequent, and *T. afroharzianum*²¹—a widely deployed biocontrol species—has been placed on the EPPO Alert List²². Earlier cautions, including those by J. L. Ricard in the late 1990s, that the benefits of *Trichoderma* should be balanced against immunological and pathogenic risks²³ underscore the timeliness of our approach.

In line with this, we refined the framing and title to foreground the phenogenomic contribution. The previous title (*Ecological Genomic Research in Trichoderma Unveils Risks and Opportunities for Its Use in Sustainable Agriculture*) placed disproportionate emphasis on biosafety, which risked obscuring the broader scientific contribution. The revised title—*Phenogenomic analysis of Trichoderma reveals determinants of fungal fitness*—foregrounds the primary strength of the work, while biosafety considerations remain integrated as one application of the phenogenomic framework, supported by the dedicated risk-assessment schema in Extended Data Fig. 6. We believe these changes improve clarity, balance, and impact.

Finally, the manuscript is currently too long, feels meandering, and also contains multiple typos that should be fixed prior to a new submission.

Response:

The revised manuscript has been substantially reduced and streamlined: redundancies have been removed, the Results and Discussion sharpened, and the narrative tightened around the ecological, evolutionary, and

biosecurity implications. We have also carefully corrected typographical errors throughout.

The authors can be congratulated for having produced a very extensive dataset and I am sure that a more streamlined version of their work will be impactful.

Response:

We are grateful for the referee's generous assessment. In revision, we have aimed to preserve the richness of the dataset while presenting it in a more concise and focused manner, and we hope that the streamlined version will indeed achieve the impact envisioned.

References

1. McGary, K. L. *et al.* Systematic discovery of nonobvious human disease models through orthologous phenotypes. *Proc. Natl. Acad. Sci. U. S. A.* **107**, 6544–6549 (2010).
2. Druzhinina, I. S. *et al.* Alternative reproductive strategies of *Hypocrea orientalis* and genetically close but clonal *Trichoderma longibrachiatum*, both capable of causing invasive mycoses of humans. *Microbiology (Reading, Engl.)* **154**, 3447–3459 (2008).
3. Samuels, G. J. *et al.* *Trichoderma asperellum* sensu lato consists of two cryptic species. *Mycologia* **102**, 944–966 (2010).
4. Druzhinina, I. S. *et al.* The *Trichoderma harzianum* demon: complex speciation history resulting in coexistence of hypothetical biological species, recent agamospecies and numerous relict lineages. *BMC Evol. Biol.* **10**, 94 (2010).
5. Cai, F. *et al.* Evolutionary compromises in fungal fitness: hydrophobins can hinder the adverse dispersal of conidiospores and challenge their survival. *ISME J.* **14**, 2610–2624 (2020).
6. Cai, F. & Druzhinina, I. S. In honor of John Bissett: authoritative guidelines on molecular identification of *Trichoderma*. *Fungal Divers.* **107**, 1–69 (2021).
7. Vimr, E. R., Kalivoda, K. A., Deszo, E. L. & Steenbergen, S. M. Diversity of microbial sialic acid metabolism. *Microbiol. Mol. Biol. Rev.* **68**, 132–153 (2004).
8. Kubicek, C. P. *et al.* Comparative genome sequence analysis underscores mycoparasitism as the ancestral life style of *Trichoderma*. *Genome Biol.* **12**, R40 (2011).
9. Kubicek, C. P. *et al.* Evolution and comparative genomics of the most common *Trichoderma* species. *BMC Genomics* **20**, 485 (2019).
10. Druzhinina, I. S. *et al.* Massive lateral transfer of genes encoding plant cell wall-degrading enzymes to the mycoparasitic fungus *Trichoderma* from its plant-associated hosts. *PLOS Genet.* **14**, e1007322 (2018).
11. Chen, S. *et al.* Genus-wide analysis of *Trichoderma* antagonism toward *Pythium* and *Globisporangium* plant pathogens and the contribution of cellulases to the antagonism. *Appl. Environ. Microbiol.* **90**, e00681-24 (2024).
12. Cao, Q. *et al.* Survival dynamics of *Trichoderma longibrachiatum* Tr58 in conidia- and chlamydospore-amended soils with different moisture levels. *Agriculture* **13**, 238 (2023).
13. Druzhinina, I. S. *et al.* *Trichoderma*: the genomics of opportunistic success. *Nat. Rev. Microbiol.* **9**, 749–759 (2011).
14. Mukherjee, P. K. *et al.* *Trichoderma* research in the genome era. *Annu. Rev. Phytopathol.* **51**, 105–129 (2013).

15. Zhang, Y.-Q. *et al.* Antibacterial activity of peptaibols from *Trichoderma longibrachiatum* SMF2 against gram-negative *Xanthomonas oryzae* pv. *oryzae*, the causal agent of bacterial leaf blight on rice. *Front. Microbiol.* **13**, 1034779 (2022).
16. Henríquez-Urrutia, M. *et al.* Circadian oscillations in *Trichoderma atroviride* and the role of core clock components in secondary metabolism, development, and mycoparasitism against the phytopathogen *Botrytis cinerea*. *eLife* **11**, e71358 (2022).
17. Xie, L. *et al.* Harzianic acid from *Trichoderma afroharzianum* is a natural product inhibitor of acetohydroxyacid synthase. *J. Am. Chem. Soc.* **143**, 9575–9584 (2021).
18. Pang, G. *et al.* Characterization of an exceptional fungal mutant enables the discovery of the specific regulator of a silent PKS–NRPS hybrid biosynthetic pathway. *J. Agric. Food Chem.* **70**, 11769–11781 (2022).
19. Nielsen, K. F., Gräfenhan, T., Zafari, D. & Thrane, U. Trichothecene production by *Trichoderma brevicompactum*. *J. Agric. Food Chem.* **53**, 8190–8196 (2005).
20. Druzhinina, I. S., Kubicek, E. M. & Kubicek, C. P. Several steps of lateral gene transfer followed by events of ‘birth-and-death’ evolution shaped a fungal sorbicillinoid biosynthetic gene cluster. *BMC Evol. Biol.* **16**, 269 (2016).
21. Pfordt, A. *et al.* *Trichoderma afroharzianum* ear rot — a new disease on maize in Europe. *Front. Agron.* **2**, 547758 (2020).
22. EPPO. EPPO warning: addition of *Trichoderma afroharzianum* to the EPPO Alert List (ear rot on maize). EPPO Rep. Serv. 04-2022, 2022/087 (2022).
23. Ricard, J. & Ricard, T. The ethics of biofungicides – A case study: *Trichoderma harzianum* ATCC 20476 on Elsanta strawberries against *Botrytis cinerea* (gray mold). *Agric. Hum. Values* **14**, 251–258 (1997).

Reviewer #1 (Remarks to the Author):

The revised manuscript by Steindorff and colleagues is much improved, with my original comments addressed satisfactorily. However, some sections and figures could use additional editing to increase clarity and impact.

Response: We thank the reviewer for the positive evaluation and helpful suggestions, which we have carefully implemented.

Line 75 I would say “Results identify Trichoderma as an ancient, genetically cohesive...”

Response: Revised accordingly.

Line 133. Please define “HOGs”.

Response: Revised accordingly.

Lines 149-164. The section “The Start of Trichoderma Life Cycle...” needs an introductory sentence explaining the rationale for the presented experiments and why the phyllosphere microbial mats are important in the first place. The paragraphs seem choppy, disconnected from each other and devoid of a broader synthesis.

Response: Revised accordingly.

Line 170. Please explain the term “idioadaptation”.

Response: Revised accordingly.

Line 204. I do not follow the opening comment “To further assess Trichoderma fitness...” Do the authors mean “fitness-related traits”?

Response: Revised accordingly.

Lines 300-301. I would say “Ecological Adaptations in Trichoderma Genomes Mediated by Regulatory Genes, Small Secreted Proteins, and Genes of Unknown Function.”

Response: Revised accordingly.

Line 302. I would say “We demonstrated that Trichoderma spp. maintain genomic cohesion while undergoing adaptive radiation and developing significant ecophysiological variability...”

Response: Revised accordingly.

Figure 2. I found panel (a) not very informative and would suggest showing only one species per strategy. The combination of panel (b) and panel (c) is great. However, I do not see the colors mentioned in the caption to be represented in panel (c). Please adjust the diagram in panel (c) to represent these colors.

Response: Revised accordingly. Thank you.

Figure 3. I love the info content of this figure. However, I would suggest using more colors and shades to denote responses on different substrates in (a). Also, using shades of purple and green for both taxonomy and responses is confusing.

Response: We thank the reviewer for this helpful suggestion. We fully agree that colours should be functionally distinct, and we have revised the figure so that taxonomic clades and trait responses use clearly separated palettes. As noted in the editor's instructions, the production team will further optimise colour shades and contrasts during figure processing to ensure the highest clarity in print.

Supplementary Figure 4, which provides support for most of the section "Profound Edaphic Incompetence...", conveys a massive amount of information. Please provide a more detailed and meaningful caption describing this content.

Response: Revised accordingly. Thank you.

Reviewer #2 (Remarks to the Author):

All my concerns have been satisfactorily addressed

Response: Thank you.

Reviewer #3 (Remarks to the Author):

I thank the authors for addressing my comments, particularly regarding the accessibility of the code used in to process the data presented in the manuscript. I still have some remarks regarding a few points in the revised manuscript.

Response: We thank the reviewer for the positive feedback and for the additional constructive remarks, which we have addressed in the revision.

I appreciate that the authors have now referenced Fig. 3b and clarified its intended contrast with phenotypic variability in Fig. 3a. However, the integration remains largely descriptive, and the term "genome inventories" overgeneralizes what is shown (specific gene category counts rather than total genomic content). To justify inclusion of panel b in the main figure, the authors should provide a minimal analytical or quantitative link, such as a measure of variance in these genomic features, a comparison of their range relative to phenotypic variation or a statement on whether any of these categories correlate with fitness metrics.

If such analysis is beyond the present scope, I recommend moving panel b to the Extended Data or omitting it altogether, since its current treatment does not meaningfully advance the results or interpretation.

Response: Thank you. We agree that this panel is primarily descriptive and made the main figure overly dense. In line with the reviewer’s recommendation—and to avoid extending the scope with additional analyses—we have moved the former panel 3b to Extended Data Fig. 6 and refined the terminology in the Results to avoid overgeneralisation. These adjustments improve clarity while leaving the conclusions unchanged.

As a minor note, I noticed some inconsistency between the original and revised Figure 1:

T. inhamatum, T. spirale and Clonostachys have different substrates between the two figures. In the revised version, T. inhamatum does not have “soil saprotrophy”, T. spirale has the symbol for ‘endophyte, epiphyte, parasite, rhizosphere colonizer’, which was absent in the first version, and Clonostachys has two new symbols for ‘mycoparasite’ and ‘litter decomposition’.

Response: We thank the reviewer for noting these differences. The apparent discrepancies were largely due to imperfect symbol alignment, which we have now corrected. During this revision, we also noted and fixed the missing second “l” in *asperelloides* across all figures. We are grateful to the reviewer for the meticulous examination of these details.

Reviewer #4 (Remarks to the Author):

This manuscript provides a valuable and highly ambitious study, integrating comparative genomics with large-scale ecophysiological, reproductive, stress-tolerance, and biotic-interaction assays across 37 strains. It has the potential to become a landmark reference on Trichoderma biology. The study also highlights biosafety concerns regarding opportunistic species, such as those with human or plant pathogenicity, especially in the context of their use as biocontrol agents or biofungicides in agriculture. I congratulate the authors for the extensive analyses conducted and the formidable dataset generated and presented. The authors have now streamlined the narrative, compared to an earlier version, and provided significant clarifications in response to earlier critiques. There is still some overinterpretation in some sections that could be tempered down.

Response: We thank the reviewer for the encouraging assessment and constructive guidance, which we have addressed in the revised manuscript.

Major comments:

The authors state that contrasting phenotypic profiles among sympatric or closely related taxa indicate character displacement. However, true character displacement would require evidence of niche partitioning driven by interspecific competition within sympatry. Here, sympatry is assumed but not demonstrated (e.g., no biogeographic co-occurrence data). Differences in BIOLOG phenotypes alone are insufficient.

Perhaps the claims could be rephrased as "consistent with" or "suggestive of" character displacement.

Response: Revised accordingly.

The authors highlight *T. afroharzianum* and *T. longibrachiatum* as high-concern species from a biosafety perspective, referencing plant and human pathogenicity. However, strain-level variation could be important and should probably be considered instead of making species-level generalizations. Agronomic recommendations rely on regulatory guidance that would focus on a strain that would be submitted to the regulatory process. Species are not regulated, but strains are. I think that the biosafety section is important and provides a solid application of the findings of this work, but it would be prudent to emphasize that the data could constitute a first-tier screening, not prescriptive regulatory conclusions.

Response: We thank the reviewer for this important point. We fully agree that regulatory decisions for microbial biological control agents are made at the strain level, and we have revised the text accordingly. Our intention was not to imply species-level regulatory conclusions but to highlight lineages that consistently exhibit high-concern phenotypes and have been associated with pathogenicity in the literature. We now explicitly frame our biosafety analysis as a first-tier screening framework, noting that strain-resolved assessment is required for agronomic use. To strengthen this context, we also note that *T. afroharzianum* has recently been placed on the EPPO Alert List, highlighting emerging phytosanitary concerns while remaining consistent with the strain-based nature of regulatory evaluation.

The authors addressed the rest of the comments appropriately in the manuscript and the rebuttal letter.